# Blow Fly (Diptera: Calliphoridae) Community Composition Across the Georgia Fall Line During Seasonal Transitions

**DOI:** 10.3390/insects16111124

**Published:** 2025-11-03

**Authors:** Edward B. Mondor, Gillian L. Johnson, Summer J. Williams, Evan C. Lampert

**Affiliations:** 1Department of Biology, Georgia Southern University, Statesboro, GA 30458, USA; 2Institute for Health Logistics & Analytics, Georgia Southern University, Statesboro, GA 30458, USA; 3Department of Biology, University of North Georgia, Oakwood, GA 30566, USA; evan.lampert@ung.edu

**Keywords:** calliphorid, entomology, forensic entomology, GLM, nMDS, species distributions, thermal biology

## Abstract

**Simple Summary:**

This study examines blow fly (Diptera: Calliphoridae) communities across the Georgia Fall Line (GFL), a geologic boundary with a distinct elevation gradient, across winter/spring and summer/fall transitions, highlighting implications for forensic entomology. By deploying baited traps at two locations north and south of the GFL, we show that blow fly community composition differed by regional biogeography more than immediate weather conditions. These findings challenge simplistic summer- vs. winter-active blow fly categorizations and underscores the need for more nuanced, geographically informed models in forensic casework.

**Abstract:**

Forensic entomologists use insect development, especially in blow flies (Diptera: Calliphoridae), to estimate the minimum postmortem interval (mPMI). Since insect activity is driven mainly by temperature, understanding geographic and seasonal variation in community composition is critical. In the southeastern United States, approximately 10 blow fly species dominate, generally classified as “summer-active” or “winter-active” flies. We studied their presence and abundance during winter/spring and summer/fall transitions across the Georgia Fall Line (GFL), a major geophysical boundary separating the Piedmont and Coastal Plain. Here we show that community structure was shaped more by regional biogeography and seasonal transitions, than by current temperature. Three species; *Calliphora livida*, *Lucilia coeruleiviridis*, and *Cochliomyia macellaria* accounted for over 70% of seasonal variation. Fly communities differed sharply across the GFL and shifted between seasonal transitions. Recognizing these geographic and temporal patterns can help forensic entomologists produce more accurate mPMI estimates in death investigations.

## 1. Introduction

Blow flies are an important study organism due to their significance in forensic investigations. Insects can be used to estimate the minimum post-mortem interval (mPMI), the minimum time since death, based on time of colonization (TOC) of the remains [1,2]. Entomologists frequently generate mPMI estimates from the TOC because insects, normally blow flies (Diptera: Calliphoridae), often arrive at a body almost immediately after death [1,3]. Blow flies, like most other insects of forensic importance, progress through all their immature stages on vertebrate remains. Temperature is the most significant factor influencing insect development; higher temperatures generally accelerate insect growth, while lower temperatures slow it down [1]. As insect development is entirely temperature-dependent, entomologists can use these poikilothermic organisms as “biological clocks”.

In the southeastern United States, there are approximately 10 common blow fly species [2]. These flies are often categorized as “summer-active” and “winter-active” [4]. Summer-active species include *Cochliomyia macellaria* (Fabricius, 1794) (secondary screwworm), *Chrysomya megacephala* (Fabricius, 1794) (Oriental latrine fly), *Chrysomya rufifacies* (Macquart, 1842) (hairy maggot blowfly), and *Lucilia cuprina* (Wiedemann, 1830) (Australian sheep blowfly). Winter-active species generally include *Calliphora livida* D.G. Hall, 1948, *Calliphora vicina* Robineau-Desvoidy, 1830 (blue bottle flies), *Cynomya cadaverina* Robineau-Desvoidy, 1830 (shiny blue bottle fly), and *Phormia regina* Meigen, 1826 (black blow fly) [4,5,6,7,8,9]. *Lucilia coeruleiviridis* Macquart, 1855 (green bottle fly), which dominates during the summer in the southeast but is unique in that forensic investigators might collect it during all seasons with temperatures exceeding its minimum temperature thresholds, has been categorized as both a summer-active and winter-active species [4,9]. These categories are useful because blow fly communities appear to change with seasonal temperature variations. Different blow fly species are therefore going to be forensically relevant in different seasons, with non-native species being especially important (e.g., they could indicate the remains were moved).

Blow fly community structure in different seasons has been well studied [10,11,12,13,14], but how community composition changes as a result of seasonal temperature increases and decreases remains less well understood. Entomologists have long known that larval development has minimum temperature thresholds, and adult activity is also temperature-dependent [4,11,13]. Adult behaviors such as flying, feeding, and ovipositing have minimum threshold temperatures that vary between species. For example, a winter-active fly like *Ca. vicina* can oviposit at temperatures as low as 10 °C [15], whereas a summer-active species like *Ch. megacephala* prefers temperatures above 15 °C for oviposition [5]. Each fly species also has a maximum temperature limit for these activities. For instance, *Ca. vicina* does not oviposit at temperatures exceeding 35 °C, while *Ch. megacephala* can tolerate and oviposit at temperatures above 40 °C [6,15]. A winter-active fly might become less active as temperatures increase in spring and start becoming more active, and thus more important to forensic investigations, as temperatures decrease in fall.

Insect communities can also vary depending on geography [16]. In the US state of Georgia, there is a geologic boundary known as the Georgia Fall Line (GFL). The GFL runs from Columbus, GA to Augusta, GA, acting as a major physiographic and ecological boundary separating the Piedmont (in the north) from the Coastal Plain region (in the south). The GFL represents the remnants of the prehistoric shoreline of the Atlantic Ocean and features a steep elevation change, resulting in a temperature gradient throughout the state. As such, it is likely that insect communities vary across the GFL [16], particularly during periods of thermal shift, i.e., winter/spring and summer/fall.

Blow fly temperature preferences directly influence the communities of insects studied by forensic entomologists [6]. For example, *Ph. regina* is predominantly summer-active in the northeastern US where summer temperatures are cooler, but is primarily winter-active in the southeastern US [9,17] which has milder temperatures in the winter. Adding additional complexity, however, is that elevation changes across even small latitudinal gradients, such as across the GFL, may create temperature differences. Climatic differences across the GFL contribute to physiographic differences such as landscape composition, including forest and soil types; the GFL forms the border between the Piedmont and Southeastern Plains US Level III ecoregions [18]. As previous research has indicated that blow fly communities differ across physiographic boundaries [19], different communities may be present on the two sides of the GFL [20]. Here we aim to determine blow fly community structure during winter/spring and summer/fall temperature transitions across the GFL.

We hypothesize that in spring there will be a greater abundance of winter-active flies in the Piedmont ecoregion (north of the GFL) as they are adapted to lower temperatures. We further hypothesize that in the fall there will be more summer-active flies in the Coastal Plain ecoregion (south of the GFL) because of the higher temperatures. These hypotheses are based on accumulated degree days (ADD) observed in North and South GA. As a result, biogeography and climatic conditions are likely to have a direct and noticeable impact on blow fly community composition.

In sum, this study addresses the following knowledge gaps. Although blow fly development has been extensively studied, particularly in relation to temperature, much less is known about how community composition changes during seasonal transitions and across physiographic boundaries. The GFL represents an ecological divide that may strongly influence insect communities, yet its role in shaping blow fly assemblages remains unclear. In addition, while forensic entomologists frequently categorize species as summer-active or winter-active, the validity of this distinction has not been well tested in the field, and some species such as *Lu. coeruleiviridis* appear to deviate from these categories. Addressing these gaps through multi-season sampling across the GFL provides an opportunity to better understand blow fly activity patterns and improve the accuracy of mPMI estimates in forensic investigations.

## 2. Materials and Methods

### 2.1. Field Sites

Field studies were conducted at two locations on opposite sides of the GFL. Each location had two spatial replicates. One location was in the Piedmont ecoregion, north of the GFL (replicate 1: 34°14′36 N, 83°51′39″ W; replicate 2: 34°14′35″ N, 83°51′58″ W). The North Georgia location was located on the University of North Georgia’s Oakwood, GA campus, in a mixed forest comprised primarily of *Pinus taeda* (loblolly pine), *Quercus* spp. (oaks), *Carya* spp. (hickories), and *Ligustrum sinense* (Chinese privet). Replicates were located approximately 300 m apart. The other location was in the Coastal Plain ecoregion south of the GFL (replicate 1: 32°25′47.414″ N, 81°47′0.594″ W; replicate 2: 32°25′5.852″ N, 81°47′23.523″ W). The South Georgia location was located on Georgia Southern University’s Statesboro, GA campus. The overstory is dominated by *Pinus palustris* (longleaf pine) and understory dominated by *Li. sinense* and *Eupatorium capillifolium* (dogfennel), particularly along the forest edges. Replicates were located approximately 1500 m apart. The two locations are separated by approximately 325 km and differ in elevation by approximately 275 m (i.e., Statesboro, GA—78 m, Oakwood, GA—353 m, approximate elevations, respectively).

### 2.2. Trapping

Five fly traps were deployed along ~50 m transects at each location and replicate. Each trap consisted of a 2 L polyethylene terephthalate (PET) soda bottle hung upside down, with small holes cut in the sides of the bottle, baited with a single chicken leg (drumstick) suspended by wire inside the bottle. Chicken legs were purchased from local supermarkets and refrigerated in the original packaging until use. This 2 L bottle is attached, using a plastic “tornado in a bottle” vortex bottle connector, to a 500 mL soda bottle oriented right side up, containing approximately 250 mL of 75% ethanol. Flies enter small holes in the 2 L bottle to get to the bait, cannot find a way out, fall to the bottom of the bottle, enter the 500 mL bottle, and are preserved in ethanol [21] (Figure 1). This trap design is an affordable modification of the baited cone traps widely used to collect flies.

Traps were placed on each side of the GFL (in the Piedmont and the Coastal plain) on the same dates; 5–10 March 2023 (winter/spring transition sampling—10–15 days from the vernal equinox) and 14–21 October 2023 (summer/fall transition sampling—21–28 days from the autumnal equinox). Traps were suspended from tree branches approximately 1.5 m off the ground, to prevent scavenging, and were located at least 10 m from other traps, to minimize odor diffusion between traps. Trap captures were only collected once, at the end of the 5 or 7 days, respectively. Different sampling intervals were required due to avoiding adverse weather conditions. Decomposition of the bait did not differ noticeably in the two replicates. Temperatures and accumulated degree days (ADD) for each location were obtained from the Georgia Environmental Monitoring Network [22]. Daily minimum and maximum temperatures were obtained, along with using the online ADD calculator to determine the ADD from 1 January to the start of each trial (spring and fall), and the ADD during each trial, using a base temperature of 10 °C (a developmental minimum temperature for many species of Calliphoridae) [23].

### 2.3. Insect Identification

Insects from each trap were sorted and placed in fresh 75% ethanol for preservation. Adult blow flies were examined under a dissecting microscope and identified to species, through morphological characters, using an online dichotomous key [24].

### 2.4. Statistical Analyses

As traps were deployed for different numbers of days in winter/spring (5 days) and summer/fall (7 days), the number of individuals of each blow fly species was analyzed as captures/day for all analyses, so they could be directly compared. Preliminary analyses showed no differences between replicated sites at each location, so this variable was eliminated from all analyses, and replicated sites were treated statistically as replicates.

The number of flies captured were analyzed with a GLM using JMP 17.0.0 [25], due to non-normal distributions in capture rates. The count data followed a Poisson distribution [26]. Independent variables in the analyses were: location (north vs. south of the GFL), sampling time (spring vs. fall), and the first-order interaction. The dependent variable was number of flies captured/day. The analysis was run with a Poisson distribution, with a log function. Overdispersion tests and intervals, as well as Firth-bias adjusted estimates, were also selected. Zero-inflation was not directly assessed in our statistical analyses. The statistics package we used does not have a specific test for zero-inflation when using GLMs. Exploratory analyses, however, indicated that we had reduced numbers of zeros by analyzing mean captures, rather than daily captures. A separate analysis was conducted for each blow fly species. Post hoc tests were conducted using the “contrasts” option in the GLM platform. *Phormia regina* and *Cy. cadaverina* were collected in such low numbers (<5 flies/sampling day), they were excluded from GLM analyses due to the large number of zeros.

Frequency counts of each species were analyzed using a series of 2 * 2 * 2 contingency tables, in which location and sampling time were treated as rows and counts were treated as columns. The Crosstabs procedure in SPSS v. 28 (IBM, Armonk, NY, USA) was used for all analyses.

Fly communities were compared between locations and sampling times using non-metric dimensional scaling (nMDS) and analysis of similarity (ANOSIM). PRIMER-7 (Primer-e, Albany, Aukland, New Zealand) was used for all multivariate analyses. nMDS was selected as the ordination technique due to zeros in the dataset, i.e., not all species were present in all traps. The number of flies captured were standardized to relative abundances and square-root transformed prior to analysis. Bray–Curtis dissimilarities were used as the measure of resemblance between samples. Once the nMDS plot was generated, resemblance levels of 25, 50, and 75 were overlaid over the samples. Similarity percentage (SIMPER) analysis was used to identify the species that contributed most to the dissimilarities.

Two one-way ANOSIM were used to compare the effects of location and sampling time on fly communities. Again, Bray–Curtis dissimilarities were treated as input values of the distance matrix. Significance of the R-statistic was assessed using 999 permutations of group membership.

## 3. Results

Mean temperatures north and south of the GFL during the winter/spring sampling period were 14.42 ± 0.88 °C and 17.65 ± 1.39 °C, respectively. During the summer/fall sampling period, the north and south average temperatures were 14.91 ± 1.00 °C and 17.05 ± 0.85 °C, respectively. As such, there were no differences between mean temperatures during winter/spring or summer/fall in the north (t_10_ = −0.37, *p* = 0.72) or in the south (t_10_ = 0.39, *p* = 0.71). There was a significant difference, however, in the mean temperatures north and south of the GFL in winter/spring (t_8_ = 1.96, *p* = 0.042), but not in summer/fall (t_12_ = 1.63, *p* = 0.13). There were large differences in accumulated degree days (ADD) between the north (winter/spring—136, summer/fall—2446) and the south (winter/spring—301, summer/fall—3163) Georgia. For the two sampling periods, there were also differences in ADD between the north (winter/spring—22, summer/fall—30) and the south (winter/spring—42, summer/fall—51).

A total of 1285 blow flies were collected during the study (n = 827 and 458 for the two sampling periods, respectively). The traps also captured other fly taxa, including 154 muscids (Diptera: Muscidae) and 41 sarcophagids (Diptera: Sarcophagidae), but those were not identified to species for this study.

### 3.1. Generalized Linear Models (GLMs)

For the three species of summer-active flies, *Co. macellaria*, *Ch. rufifacies*, and *Ch. megacephala* it was not possible to run GLMs as the only sampling period in which they were collected was south of the GFL, in the summer/fall (Figure 2). There were no captures of any of these species south of the GFL in the winter/spring, or north of the GFL in winter/spring or summer/fall.

The winter-active fly, *Ca. livida*, was more likely to be captured in the winter/spring (x^2^_1_ = 47.03, *p* < 0.0001), but captures were independent of location (x^2^_1_ = 0, *p* = 1.0), and there was no interaction between location and sampling time (x^2^_1_ = 0.0, *p* = 1.0). The same was true for *Ca. vicina*; it was more likely to be captured in the winter/spring (x^2^_1_ = 24.80, *p* < 0.0001), but not in any location (x^2^_1_ = 0.0, *p* =1.0), nor was there a significant interaction (x^2^_1_ = 0.0, *p* = 1.0) (Figure 2).

*Lucilia coeruleiviridis* was the only fly that did not match the pattern of a summer-active or winter-active fly. It was more likely to be captured in the summer/fall (x^2^_1_ = 9.33, *p* = 0.0022), marginally more likely to be captured north of the GFL (x^2^_1_ = 3.75, *p* = 0.053), and there was also a significant interaction between location and sampling time (x^2^_1_ = 46.27, *p* < 0.0001) (Figure 3).

### 3.2. Contingency Tables

The summer-active flies *Co. macellaria*, *Ch. rufifacies*, and *Ch. megacephala* were only collected south of the GFL, and all three species were more likely to be collected in the summer/fall. The winter-active flies *Ca. livida* and *Ca. vicina* were more likely to be collected in the winter/spring regardless of location. *Lucilia coeruleiviridis* was more likely to be collected in summer/fall in both locations (Table 1).

### 3.3. Non-Metric Dimensional Scaling (nMDS) and ANOSIM

nMDS revealed four groups of samples with at least 50% similarity, grouped by location and season (Figure 4). Both communities sampled south of the GFL in the summer/fall clustered at 25% similarity with communities sampled north of the GFL in the summer/fall. The communities sampled north of the GFL in the winter/spring were the most unique communities, as they were the only communities that failed to cluster with other communities at 25% similarity.

ANOSIM revealed that the replicate had little effect on fly communities when sampling periods were combined. For the pairwise comparison between the two replicates south of the GFL, a negative R-statistic indicated greater dissimilarities between traps within each location, compared to a lower degree of dissimilarity between sites (Table 1). All other pairwise comparisons generated R < 0.15, with the greatest differences between the locations south of the GFL and the location north of the GFL. The pairwise comparisons between all sites north of the GFL and south of the GFL had R-statistics close to 0, indicating that each group had a mixture of high and low ranked distances. According to ANOSIM, sampling time had a substantial effect on fly communities, with dissimilar communities collected in winter/spring and summer/fall (Table 2).

SIMPER analysis was performed only to compare winter/spring and summer/fall because there were fewer differences in fly communities among sites. Three taxa contributed to >70% of the dissimilarities between fly communities in the winter/spring and summer/fall. *Calliphora livida* was collected only in the winter/spring, and *Co. macellaria* was collected only in the summer/fall. Similarly, *Lu. coeruleiviridis* was more abundant in summer/fall (Table 3).

## 4. Discussion

Determinants of blow fly community assemblages has long been a subject of debate [4,27,28,29]. In this project we sought to determine the composition of blow fly communities during winter/spring and summer/fall transitions across the GFL. As blow fly communities were very different north of the GFL and south of the GFL, even though temperatures between the spring and fall samplings in the north, as well as in the south, were not significantly different, these differences are believed to have been largely influenced by regional biogeographic factors such as elevation, vegetation, and soil composition, in addition to temperature. Furthermore, our results indicate that there is some validity to blow flies being categorized as summer-active and winter active species.

Immediate weather conditions (e.g., temperatures) were not the primary factor in fly captures in our study. Temperatures at each location (north and south) differed by less than 1 °C between winter/spring and summer/fall, while the temperature difference between the two locations was approximately 2.5 °C. The ADD, however, which plays a significant role in seasonal effects, varied considerably. For example, the winter/spring ADD was more than twice as large south of the GFL. Previous research suggests that ADD, rather than immediate temperature, is a key factor in structuring blow fly communities [12]. This finding is consistent with the observation that many insects have seasonal preferences determined by their optimal activity temperatures [4,11,13]. This raises an important question: in regions where flies do not require diapause, what primarily determines their abundance?

We hypothesized that summer-active flies such as *Co. macellaria*, *Ch. rufifacies*, and *Ch. megacephala* would be more abundant in the summer/fall in the south due to the higher accumulated degree days. This hypothesis was supported; in fact, all three summer-active species were only collected south of the GFL. All three summer-active species occur north of the GFL during peak summer, and both *Co. macellaria* and *Ch. rufifacies* have been documented at the same location sampled here [27]; however, in this study they were either inactive near the northern sampling sites or just not present at this time of year. It is also plausible that these species occupied distinct microhabitats (e.g., rural versus forested environments) during specific seasonal periods, a hypothesis that warrants further investigation through targeted experimental studies. Likewise, we hypothesized that winter-active flies such as *Ca. livida* and *Ca. vicina* would be more likely to be captured north of the GFL in the winter/spring, due to the lower accumulated degree days. This hypothesis was also supported. These two species were also collected south of the GFL in the winter/spring, albeit in lower numbers compared to north of the GFL. Populations of these winter-active species south of the GFL were possibly less active by March as they had experienced greater ADD compared to more northern populations.

Some species, such as *Cy. cadaverina*, *Ph. regina*, and *Lu. coeruleiviridis* did not, however, fit our hypotheses. It is very likely that *Cy. cadaverina* and *Lu. coeruleiviridis* cannot be distinctly categorized as summer-active and winter-active species. *Cyanomya cadaverina* appears to prefer moderate temperatures, like *Cy. mortuorum*, a European species with similar physiology, which does not show a strong preference for extreme heat or cold, making it difficult to categorize as a summer-active or winter-active species [30]. Likewise, *Lu. coeruleiviridis* is prevalent year-round in the southeastern US and is one of the most abundant flies in this region [3,31]. Its reduced abundance during our study may be due to various factors. One possibility is that other summer-active flies are outcompeting this previously dominant species, particularly *Ch. megacephala*, an invasive species [32,33]. Alternatively, *Lu. coeruleiviridis* might peak in abundance in a different month, perhaps filling a niche during the extreme summer months when temperatures are higher. In some summer studies, this species can make up over 60% of trap captures [34,35]. Further behavioral studies investigating the activity patterns of *Cy. cadaverina* and *Lu. coeruleiviridis* at low, moderate, and higher temperatures can help researchers better understand their thermal flexibilities.

*Phormia regina* was a notable outlier in our study. While this fly is typically classified as a winter-active species [9,17], we found it in the northern region during the summer/fall. This anomaly, however, may be explained by the migratory nature of *Ph. regina*, which spends summers in the northeastern US and winters in the south [36]. It is possible that our study captured this species in north Georgia during the early stages of its southward migration for winter, accounting for its presence in low numbers. Future studies using isotopic analysis or population genetic structure could verify this.

In this study, we analyzed fly captures from two locations in Georgia separated by approximately 2° of latitude. Even over this small geographic distance, we observed striking differences in fly communities. In general, distinct communities were present north and south of the GFL across winter/spring and summer/fall transitions. Over 70% of the seasonal variation was attributable to just three fly species. The winter-active fly *Ca. vicina*, which was collected only in the winter/spring, contributed most to the community differences. Since *Ca. vicina* was the most common species collected in the winter/spring, this species might be an ecologically significant contributor to nutrient cycles at that time of year. *Lucilia coeruleiviridis*, which was common in both winter/spring and summer/fall, was the second most important contributor to the community differences. The final species contributing significantly to community differences was *Co. macellaria*. While *Co. macellaria* is a summer-active species, its abundance in the summer/fall suggests that it might be ecologically significant as temperatures cool during the summer/fall transition.

## 5. Conclusions

A better understanding of blow fly community composition can enhance forensic entomology by identifying shifts in insect succession patterns [4]. By accounting for seasonal transitions and geophysical boundaries, forensic entomologists can improve the accuracy of mPMI estimates by clarifying which blow fly species are likely to be active at a given time and place [1]. Seasonal shifts determine when species increase or decrease, while geographic boundaries can influence local species distributions. Because different blow fly species develop at different rates, applying the correct developmental data depends on recognizing temporal and geographic patterns of species distributions. Incorporating this knowledge allows forensic entomologists to refine species identification, reduce uncertainty, and generate more precise mPMI estimates. This information could also alert investigators to variables they might not otherwise consider. For instance, if a body found in the Piedmont (north of the GFL) in October has large numbers of *Co. macellaria*, it may indicate that the remains originated from a more southern region within the state. Future studies should focus on monthly (or seasonal) sampling over multiple years, over elevational (and latitudinal) gradients across the US, and internationally [37].

## Figures and Tables

**Figure 1 insects-16-01124-f001:**
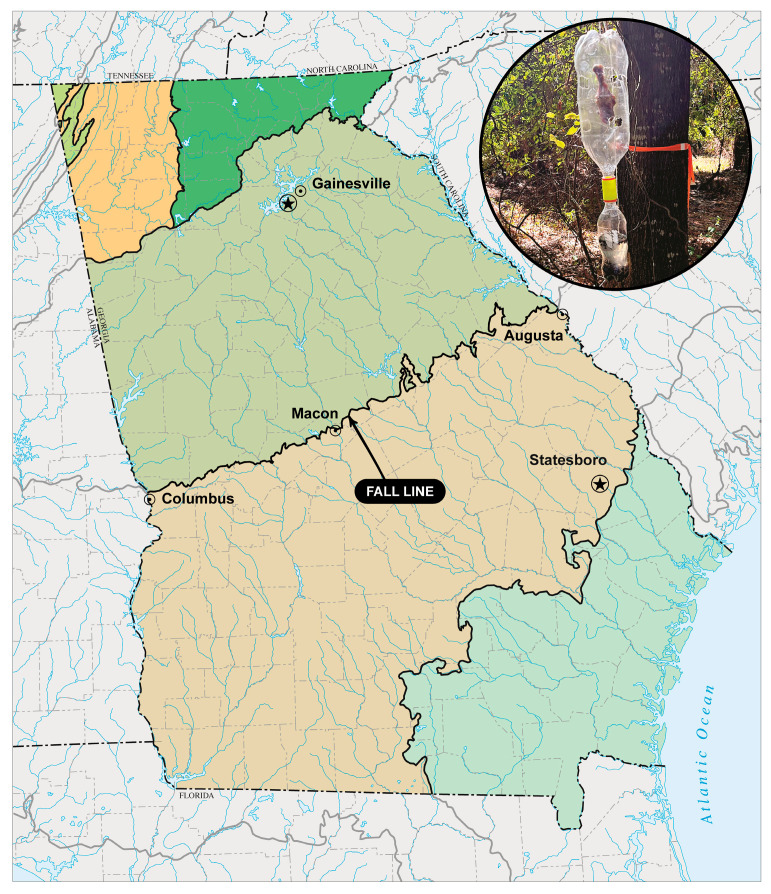
Sampling locations (indicated by stars) relative to the Georgia Fall Line (GFL). Map downloaded and adapted from the USA EPA Level 3 Ecoregions website [18]. Inset shows the bottle trap design; photograph was taken during the summer/fall transition sampling period.

**Figure 2 insects-16-01124-f002:**
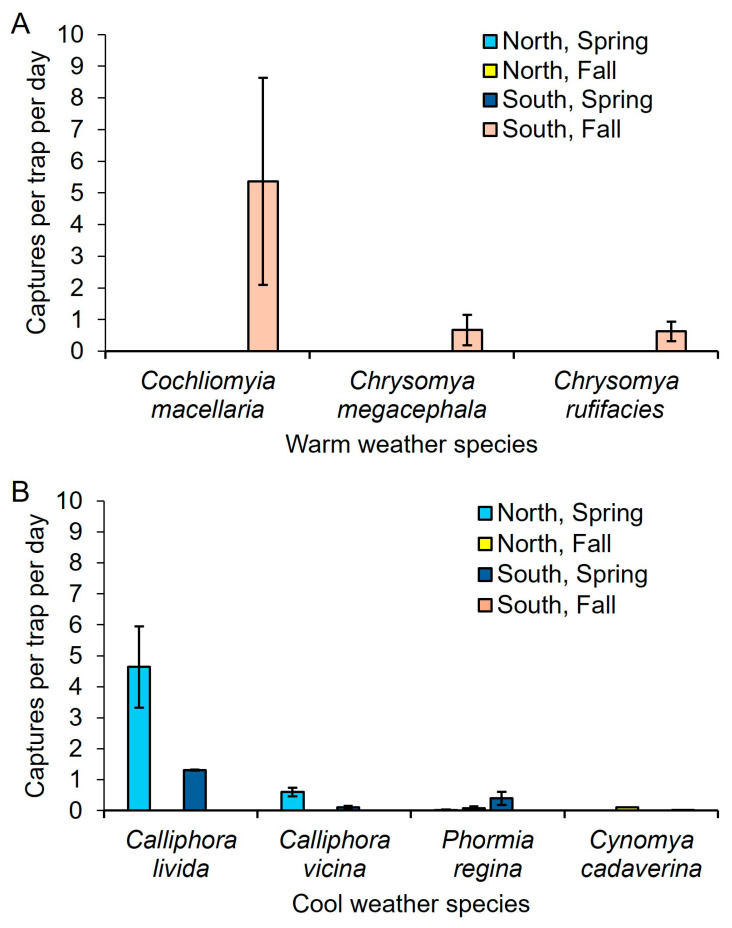
Numbers of (**A**) summer-active and (**B**) winter-active species captured per trap each day, North and South of the GFL during the winter/spring and summer/fall transitions. *Cochliomyia macellaria*, *Chrysomya megacephala*, and *Chrysomya rufifacies* were only captured South of the GFL in the summer/fall transition. *Phormia regina* and *Cynomya cadaverina* were not included in GLM statistical analyses due to low trap catches.

**Figure 3 insects-16-01124-f003:**
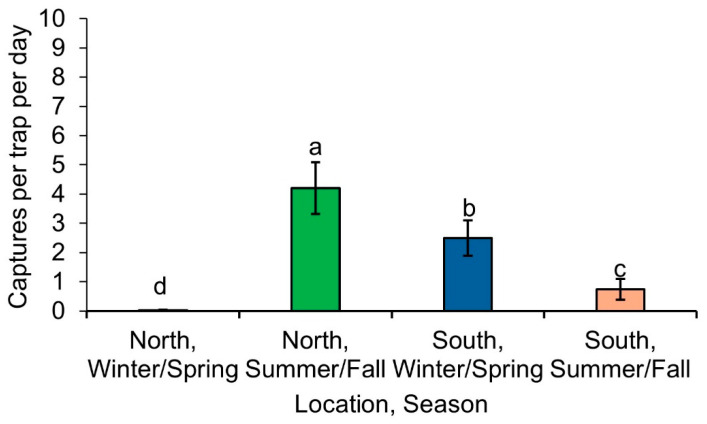
Numbers of *Lucilia coeruleiviridis* captured per trap each day, North and South of the GFL, during winter/spring and summer/fall transitions. Different letters over columns indicate significant differences (*p* < 0.05).

**Figure 4 insects-16-01124-f004:**
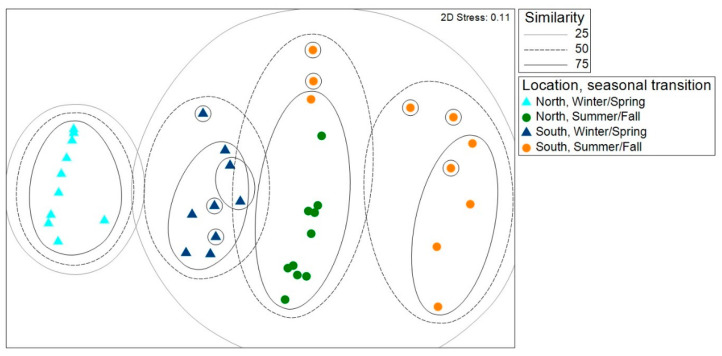
Non-metric multidimensional scaling (nMDS) plot of blow fly communities North and South of the GFL in winter/spring and summer/fall transitions. Triangles represent the winter/spring transition and circles represent the summer/fall transition. Light blue and green symbols represent collections North of the GFL, and dark blue and orange symbols represent collections South of the GFL.

**Table 1 insects-16-01124-t001:** Comparison of chances of collecting each blow fly species North and South of the GFL during the winter/spring and summer/fall seasonal transitions. The “Seasonal transition” column indicates the seasonal transition in which that blow fly species was more likely to be collected.

Species	Location	Seasonal Transition		
			χ^2^	*p*
Summer-active flies				
*Cochliomyia macellaria*	North	Not collected	-	-
	South	Summer/fall	9.98	0.002
*Chrysomya rufifacies*	North	Not collected	-	-
	South	Summer/fall	6.11	0.01
*Chrysomya megacephala*	North	Not collected	-	-
	South	Summer/fall	6.11	0.01
Winter-active flies				
*Cynomya cadaverina*	North	Summer/fall	8.57	0.003
	South	No difference	0.95	0.33
*Phormia regina*	North	No difference	1.25	0.26
	South	Winter/spring	9.74	0.002
*Calliphora livida*	North	Winter/spring	20.00	<0.001
	South	Winter/spring	19.00	<0.001
*Calliphora vicina*	North	Winter/spring	20.00	<0.001
	South	Winter/spring	3.96	0.047
Summer- and winter-active flies				
*Lucilia coeruleiviridis*	North	Winter/fall	16.36	<0.001
	South	Collected in both	-	-

**Table 2 insects-16-01124-t002:** Analysis of similarity (ANOSIM) results. The R-statistic ranges from −1 to +1. A value approaching +1 indicates the samples are highly similar within groups, and highly dissimilar between groups.

Comparison	R	*p*-Value
Location (North vs. South)		
Winter/spring transition	0.91	0.001
Summer/fall transition	0.68	0.001
Seasonal transition (Winter/spring vs. summer/fall)		
South	0.75	0.001
North	1.00	0.001
South winter/spring transition, North summer/fall transition	0.70	0.001
North winter/spring transition, South summer/fall transition	0.99	0.001

**Table 3 insects-16-01124-t003:** Similarity percentage (SIMPER) comparisons of spring and fall collections. “Contribution %” is the blow fly species percent contribution to the Bray–Curtis dissimilarities between winter/spring transition and summer/fall transition collections. “Cumulative %” refers to the cumulative percentage of the most important fly species to dissimilarities between the two seasonal transitions. Blow fly species are listed in decreasing order of importance to Bray–Curtis dissimilarities.

Species	Contribution %	Cumulative %
*Calliphora livida*	34.99	34.99
*Lucilia coeruleiviridis*	25.40	60.39
*Cochliomyia macellaria*	12.03	72.42

## Data Availability

The original contributions presented in this study are included in the article. Further inquiries can be directed to the corresponding author.

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
