# Peer review of "Blow Fly (Diptera: Calliphoridae) Community Composition Across the Georgia Fall Line During Seasonal Transitions"

_insects, 2025, doi:10.3390/insects16111124_

Round 1
Reviewer 1 Report
Comments and Suggestions for Authors
Review of Insects-3739762
Blow Fly (Diptera: Calliphoridae) Community Composition 2 Across the Georgia Fall Line: Insights from Spring and Fall 3 Sampling
The manuscript provides an overview of a very simple study to examine the blow fly species composition in two different seasons in 2 geographic locations in the same state. The authors indicate the importance of geographic and temporal distribution of blow flies and how this may influence forensic investigations involving forensically relevant blow flies. The methods could use additional information, the results are lacking and only demonstrate information for a very small number of flies overall, based on only 1 trapping event and the discussion doesn’t adequately explain how this information could benefit forensic entomology, outside of just documentation of which season these species may be present.
Different key words that are not already in the title
Page 2, lines 38-40: I don’t think this generalization is true and should be removed from the manuscript; many factors contribute to abundance and distribution and this statement isn’t an accurate description
Methods
How is season defined by the authors?
Page 3, lines 17-23: Why was a chicken leg used; was the meat fresh or aged in any way?
Page 4, lines 6-7: how far away were the traps from the weather station location?
Page 4, lines 8-9: why ADD rather than ADH?
Page 4, lines 9-10: Why was a base temperature of 10 selected?
Why weren’t data loggers placed with the traps? No mention of humidity
Results
If traps were only collected at the end of the 5-7 day period, how are there daily fly counts per trap? Authors do not provide an explanation of this and make some generalizations just by dividing the total flies by number of days.
ADD is mentioned in the methods, but the results of these ADD calculations are not provided. Is there a supplemental file with these calculations?
Were there any other families of flies in the traps? The authors don’t mention anything else that was captured.
What were the overall abundances of each species? Hard to determine from the figures and tables as to the overall counts of each species of fly
Discussion
Page 8, line 35: I think the authors are referring to the species Cyanomya mortuorum here.
How can this information be used in forensic entomology cases? There’s limited results and no clear trends that would add value for estimates of TOC. The conclusions made by the authors are based on a small number of traps in a limited area, only collected in 2 times of the year.
Reviewer 2 Report
Comments and Suggestions for Authors
Dear authors,
Thank you so much for this manuscript on blowflies. I think the study is interesting, however, I have some concerns about the methodology that has been followed as well as the usefulness of the presentation of the results for forensic entomologist. Also some improvements should be done:
- Page 1, line 20: 'to assist in determining a minimum post-mortem interval (mPMI) estimate'. This sentence is a bit confusing, please rephrase.
- Page 2, Line 9-13: you mention the most abundant species, however, the correct way to name them is not using the common name, it is the scientific name in this way: name specie authority, year (Order: Family), for example: Chrysomya megacephala (Fabricius, 1794) (Diptera: Calliphoridae), this should be done the first time that you mention the species in the text. The common names are not important in this manuscript, if anyway you want to mention them, you can do it in second place.
- Page 2, Line 17: 'How blow fly community structure shifts with temperature during the spring and fall remains poorly understood'. This is not complete true, there are a lot of studies that study this type of shifts, so please check more bibliography or rephase this sentence.
- Page 2, Line 18-21: minimum threshold temperature is not new in adults, all the stages of the life cycles has it, please rephrase this part of the paragraph. This is not something new.
- Page 2, Lines 25, 26, and page 8, line 33: you mention three species: C. vicina (genus Calliphora), C. megacephala (genus Chrysomya), and C. cadaverina (genus Cyanomya). The short for of the genus should be in different form as they are different genus, for example, C. for Calliphora, Ch. for Chrysomya, and Cy. for Cyanomya. And this should be respected throught the text.
- Page 2, Lines 31-32: this sentence should be rephrased, it is not clear for the reader.
- Page 2, Line 37: the black blow fly (Phormia regina), you already mentioned the common name and the scientific name before, no need to repeat the information, choose one.
- Page 3, Line 5: 'Field studies were conducted at four sites.' This is not true, you did your fieldwork in 2 sites only (North and South), with 2 replicates in each site. How far this two sites in each location were? What about the 5 traps?
- Page 3, Line 23: 'and are preserved in ethanol'. Which % of ethanol did you use?
- Figure 1: please improve it and take a picture with a closer angle, or make a drawing for the type of traps. This image is not clear to understand the mechanism of the trap.
- Page 4, Line 2: why the 2 seasons (March and October) had different duration (5 and 7 days9?
- Page 4, Line 8: how did you calculate the ADD? Nothing is mentioned in the methodology
- Page 4, Lines 18-19: as you mentioned, the 2 sites for each locations cannot be considered sites, you should consider them replicates, so you can rephrase the first part of the methodology and make it simple.
- Page 5, Line 2-3: 'The mean temperatures (± 1 SE) north and south of the GFL during the spring sam-2 pling period were 14.42 (± 0.88) °C and 17.65 (± 1.39) °C, respectively.' The first part of the sentence is not clear, and (± 1 SE) can be removed. I suggest to change the SE to this: 14.42 ± 0.88 °C (without parenthesis).
- Page 5, Line 14: Figure 1? I think this is not correct.
- Page 5, lines 26-27: P. regina and C. cadaverina were not included in the stats, but they are important. These species should be appeared somewhere, so a table or a graph with all the species would be useful.
- Figure 3: same with Lucilia coeruleiviridis, I don't understand why it is isolated in only one graph. If you want to see the differences in the community, you should place them all together.
- Page 8, Lines 9-13: although the geographical differences are important, there are not more important than the temperature. Then after that, you mention that there are some species classified as winter and other as summer species. This is temperature dependent. So you are trying to explain the differences of the blowflies community with the temperature anyway, maybe you need to talk more about the minimum threshold temperature that the species needs, that's why they are winter and summer species...instead to talk about the mean temperature, talk about the minimum and the maximum daily temperature. I think more justification and discussion is needed.
- Page 8, Line 16: again you mention that your result doesn't depend on the temperature, but the ADD calculations depend on temperature. So this paragraph should be re-discussed.
- Page 8, Line 41: don't capitalised oriental
- References: for the 1st reference, there is a newer edition of this book.
In general, for the methodology, it should be improved the manuscript explaining correctly, also another important thing for a forensic entomologist is the different waves of the insects, you only collect all the flies at the end of the experiment, it would be helpful to know which species arrived and was collected first.
For the results, there a lot of graphs and tables that as a forensic entomologist are not useful. Please make some table/graph with all the species.
The discussion should be improved a lot. There is too little comparison with other studies, less than 2 pages of discussion is very short. Also 31 references in a topic that has been studied a lot is too little as well.
After all these amendment, I can reconsider my decision. Thank you so much.
Reviewer 3 Report
Comments and Suggestions for Authors
This study offers a timely examination of blow fly communities across Georgia's Fall Line, effectively challenging simplified seasonal classifications in forensic entomology. Your methodological approach particularly the GLM, nMDS, ANOSIM, and SIMPER applications provides solid analytical grounding. To strengthen impact:
- Sharpen the introduction by explicitly naming the knowledge gaps your work addresses, particularly regarding transitional seasons' ecological significance in forensic contexts
- Enhance methods transparency, document site-specific environmental variables, justify unequal spring/fall sampling durations, and clarify the 10°C ADD threshold selection
- In results, probe whether observed 2-3°C differences actually impact fly behavior, and discuss sampling adequacy for rare species
- The discussion would benefit from deeper exploration of ecological drivers (competition, migration barriers, microhabitat effects) and clearer forensic applicability statements
- Consider adding a conclusion section synthesizing key implications and future research avenues
These refinements will elevate the manuscript's contribution to both ecological theory and forensic practice.

Reviewer 4 Report
Comments and Suggestions for Authors
Review of INSECTS 3739762 Blow fly (Diptera: Calliphoridae) community composition across the Georgia fall line: Insights from spring & fall sampling.
This study provides great background about fly communities in this area of Georgia. The sample size is small though, with only one time point collection for each season, which leads to concerns if these communities represent the season or just that month/time point. Overall, it is written very well. Please see below for additional comments.
Introduction
Nice background and set up for the project, I would consider including a map of Georgia showing the GFL and photos of what each habitat looks like or information about each.
Page 2 line 41, I think ‘is’ is supposed to be ‘if’
Page 2 line 45 blow fly needs to be split.
Materials & Methods
You touch upon what is in each habitat here, but a photo about location whether previous or here I think would be informative. Additionally, I would include elevation for each site.
How were your length of trap deployments determined? The 5 v 7 days, were the ADD similar between the two trapping periods (while traps were exposed)?
Why were March and October chosen to represent your seasons? I don’t believe you delineated what each season was.
Results
You state that there are summer active and winter active flies, but you don’t ever list out the 10 species. I think this would be informative for the project and overall manuscript.
For Table 1 the distinction of summer-active v winter-active is from your findings or previous literature?
Do you think an indicator species analysis would be beneficial to confirm the type of fly as far as summer or winter active in comparison to what you found in your traps?
Discussion
I think the transformation to ADD is an interesting one, but that will not parse out any extreme weather patterns. For example if there is one day with extreme heat that will increase your ADD but it may not have been prolonged enough to impact fly communities. And thinking about fly biology, how far out prior to their arrival would impact them? You discuss the presence of the summer active flies more likely in the fall due to the ADD, but your ADD values were calculated from January, do you think more recent ADD values (for example, the one month prior, or two) would be more informative? Starting from Jan 1 is a man-made construct so in theory we could use any timepoint?
Meeds et al. 2023 saw that although temperatures in spring and fall were similar the number of flies and species richness were starkly different, indicating the months leading up to those collections were more important than the temperatures themselves. This is a really interesting concept I think can be investigated further.
You discuss the summer active and winter active flies and how it is hard to categorize several species as one of those. Why are those the only two categories? I think providing a table to show what the monthly averages are like at each site would be valuable as well. Most readers will not be familiar with the temperatures of those regions and having that background would strengthen the paper.
I do not know the elevations of the sites north and south of the line but I am curious to see how they differ and if it is drastic if this is a variable that needs to be considered.
Reviewer 5 Report
Comments and Suggestions for Authors
Most of my issues are grammatical and are in the attached pdf - making sure "blow fly/ies" is two words and shortening species' names and removing common names after their first introduction.
It would be beneficial to state in the article why you chose along a forest line for your traps. Some of the species (Ch. rufifacies, for example) prefer open, sunny areas which may have impacted the numbers of each fly collected. I would be curious to see if these relationships hold with traps at the wood edge and out in the open.

Round 2
Reviewer 1 Report
Comments and Suggestions for Authors
The authors have addressed the previous comments and concerns, but the limited information would be more appropriate for a short communication
Reviewer 2 Report
Comments and Suggestions for Authors
Dear author,
Thank you for improving the manuscript. However, there is still controversy in the paper regarding your results and statements. You mentioned that you don't take into account the temperature but, as I mentioned before ADD is temperature dependent, geography and seasons also. Although, the mean temperature is only 1ºC different in both locations, if ADD is very different, and you acknowledge this, this is related to temperature. Also using the temperature and the ADD for all the season, instead only for the 5 or 7 days of sampling has a lot of impact on how your results are seen, also for using 10ºC as a minimum threshold temperature, instead looking for it for each species.
The Figure 1 is still small to see the traps. C. macellaria should be Co. macellaria. L. sinense should be Li. sinense. Check rufifacies along the text, in a lot of places is not correctly spelled.
Regarding your conclusions: 'Seasonal shifts determine when species increase or decrease, while geographic boundaries can influence local species distributions. Because different blow fly species develop at different rates, applying the correct developmental data depends on recognizing these temporal and geographic patterns. Incorporating this knowledge allows forensic entomologists to refine species identification, reduce uncertainty, and generate more precise mPMI estimates' The development data depends on temperature, not temporal and geographic patterns, and also the mPMI estimations depends on the temperature. So all the explanation through the text about the geography doesn't make sense because refusing that this doesn't depend on the temperature is not correct in forensic entomology.
The discussion is still weak. Thank you so much.
Reviewer 3 Report
Comments and Suggestions for Authors
I have reviewed the revised version of the manuscript. As suggested in the previous round, the authors have incorporated all the recommendations. In my view, the manuscript is now ready for acceptance. My recommendation at this stage is to accept.
Reviewer 4 Report
Comments and Suggestions for Authors
Insects Revision2
Introduction
A clear explanation of TOC v mPMI needs to be written. Insects are used to calculate a TOC which can be used for a mPMI (as sentence 38-39 on pg 1 states), but we do not always generate a mPMI (following sentence).
Methods
Much better explanation of trap location.
Why are March and October considered transition periods between seasons? No references/ data to support this was provided.
Results
Were preemptive ADD calculations used for length of time trap was out (ADD values in section 3)? They were larger both times in the southern area, but is a 20 ADD difference considered large between seasons or a 10 ADD between seasons?
What is meant by second replicate in paragraph 2 in results? I think it means site location, but a replicate would indicate within the same area.
Captures per day is not a true way to reflect what is occurring over a 5-7 day period. There is no way to determine when the flies arrived within that window or if one species arrived due to olfactory cues of other flies, the bait etc. Your data would be better represented as relative abundance showing total captures over the duration of your trap deployment (esp since the amount of days were different).
For 3.3 don’t you mean both of the communities in the south (not 3) clustered with 1 in the north (total 3)?
Discussion
Several claims made in this section are a bit strong based on the limited data available from this study’s findings. I would suggest reviewing this section and support with literature or reduce claims.